# Health Promotion on Instagram: Descriptive–Correlational Study and Predictive Factors of Influencers’ Content

**DOI:** 10.3390/ijerph192315817

**Published:** 2022-11-28

**Authors:** Laura Picazo-Sánchez, Rosa Domínguez-Martín, David García-Marín

**Affiliations:** 1Department of ICT Applied to Education and Media Literacy, Faculty of Education, Universidad Internacional de Valencia (VIU), 46002 Valencia, Spain; 2Department of Pedagogy, Faculty of Education, Universidad Internacional de Valencia (VIU), 46002 Valencia, Spain; 3Department of Journalism and Corporate Communication, Faculty of Communication Sciences, Universidad Rey Juan Carlos (URJC), 28933 Móstoles, Spain

**Keywords:** influencer, Instagram, health promotion, followers, social media, misleading content, media education

## Abstract

The pandemic has accentuated the power that influencers have to influence their followers. Various scientific approaches highlight the lack of moral and ethical responsibility of these creators when disseminating content under highly sensitive tags such as health. This article presents a correlational statistical study of 443 Instagram accounts with more than one million followers belonging to health-related categories. This study aims to describe the content of these profiles and their authors and to determine whether they promote health as accounts that disseminate health-related content, identifying predictive factors of their content topics. In addition, it aims to portray their followers and establish correlations between the gender of the self-described health influencers, the characteristics of their audience and the messages that these prescribers share. Health promotion is not the predominant narrative among these influencers, who tend to promote beauty and normative bodies over health matters. A correlation is observed between posting health content, the gender of the influencers and the average age of their audiences. The study concludes with a discussion on the role of public media education and the improvement of ethical protocols on social networks to limit the impact of misleading and false content on sensitive topics, increasing the influence of real health prescribers.

## 1. Introduction and Literature Review

The current social media scenario has transformed communicative, informative and social logic [1]. This digital revolution has taken hold so fast that formal education needs to be updated with the incorporation of digital competence and media and information literacy, understood as the preparation for digital life with the development of critical thinking to consume and create media products [2], as the use of digital devices and the Internet is becoming essential for adolescents, who find in them the motivation to connect on an emotional, informational and social level with other people and to explore their interests in greater depth.

According to data provided by the Qustodio annual report for 2021 [3], which measured Internet use among 5–15-year-olds in the US, UK and Spain, children spend an average of 4 h a day using online applications, with a 28% increase in time spent on communication applications (such as social networks) compared to 2020.

A decade ago, Gutiérrez and Tyner [2] already highlighted the importance of educators being familiar with social networks for them to be present and to ensure proper and critical use of them. EUKids Online 2020 points out that, in Europe, the average time a child aged between 9 and 11 spends on the Internet is 1.9 h, increasing to 3.8 h for adolescents aged between 15 and 16 [4]. Additionally, 25% of young Europeans aged 9–16 reported to have had negative experiences online [4].

The normalised use of the Internet, as well as the emergence of new capabilities and lifestyles, means that users of social networks have become either consumers or a reference for other users, mainly the millennial audience [5].

Web users encounter two situations when they perform a range of searches on social networks [6]. On the one hand, information from reliable sources and on the other, misinformation that is neither controlled nor reliable. Several studies [7,8,9] reveal a new communication ecosystem in which adolescents constantly use social networks for entertainment but also to access information, making the detection of false and malicious content essential.

In this context, Caro-Castaño [10] demonstrates the centrality of influencers as prescribers of products and practices that structure, organise and reconstruct their discourse based on the distinction between social classes. This raises questions about identity, aspirations and the effects on the complex adolescent processes of self-affirmation, identity, critical thinking and relationships with peers. Social networks play an essential role in this construction, which is not always considered in formal education.

Different authors have been stating for years that social networks are an opportunity to work on relevant aspects of students’ maturity due to their influence on them [11,12]. Along these lines, Andrade et al. [13] state that young people are growing up in a digital and media context, influenced by social networks and enhanced by online education, given the recent health events.

As stated by Feijoo and Sádaba [14], preference-based targeted advertising is widespread in the digital world, including social networks and the Internet as a whole, with children and adolescents representing a key audience for brands that seek to attract these age groups through entertainment, games and other actions that are not easily identifiable by users, since they have not been sufficiently educated to do so.

While educational institutions are gradually incorporating new technologies into the classroom, media education is still far from being understood, studied and implemented [15]. Along these lines, a study conducted by Borah and Xiao [16] already demonstrates how the health promotion discourse is conditioned by the credibility logics found in social networks (social support and likes [16]), rather than by the sources transmitting them. It also indicates the importance given to health information, which was notably in demand both during and after the COVID-19 pandemic, in which users searched for graphic and visual information specifically conveyed by these influencers [17].

These profiles (influencers) are endorsers, where endorsement is understood as the power of a media individual to recommend the use of a service, product or whatever appears on their social networks, especially Instagram [18]. This recommendation works, as users tend to show a preference for products that are used or recommended by individuals they consider of interest, which they follow on social networks [19]. For Childers, Lemon and Hoy [20], the influencer has the freedom to control the way in which they create content to present on social media, which is often seen as more authentic and engaging by the target audience than if presented directly by the brand. Influence is most effectively exerted by conveying personal opinions about brands in their posts, intervening in the behaviour of the audience, especially younger users, who trust and value their judgement when making purchasing decisions [21,22,23].

### Social Networks and Health

As noted, the pandemic highlighted the role of social networks in the dissemination of relevant messages [24,25], proving to be of great communicative utility in crisis situations, emergencies and disasters, as occurred during the COVID-19 health crisis [26]. Indeed, this health crisis led to an infodemic situation, understood as the conjunction between information and epidemic [27]. This terminology refers to the excess of information, not necessarily truthful, on a given topic. This reality has led to several solutions from different areas and perspectives, including media, digital and information education [28]. Disinformation and infodemic, especially noticeable in situations such as the COVID-19 emergency, pose complex challenges to be addressed [29], which is why it is important to consider that, some of the factors associated with the growth and spreading of the infodemic, García-Saisó et al. [30] include:Difficulty in searching for and critically selecting reliable information.Lack of judgement and tools to obtain accurate information in the right format and at the right time.Unawareness of the usage and relevance of health-related digital applications.

As stated by Herrera-Peco [6], it is becoming increasingly common for patients and relatives to inform themselves in advance and go to medical appointments with certain knowledge they have found from unreliable sources on the Internet and social networks or assuming that the information that influencers have posted on their profiles is true. It should be noted that an influencer may promote or suggest a product without providing all the necessary information, making potentially misleading recommendations that could negatively affect the consumer, intentionally or unintentionally [31]. There is no prior control of the information provided through advertising, but this does not imply that all messages or conduct regarding such medicines are authorised [32]. This surveillance already exists in the traditional media, assuming, as indicated by Picazo-Sánchez, de Frutos-Torres and Gutiérrez-Martín [33], social responsibility in the alleged veracity of the information offered by mass communication professionals, given its educational and informative purpose, which should be a priority. However, this responsibility is uncertain when it comes to the new opinion leaders in social media, which raises the need to control self-described health-related content on social media.

The network’s users who have their own profile can select topics or categories about which the content they create and share is based on, according to their own criteria. In the case of Instagram, the categories are artist, musician/music band, blogger, clothing (brand), community, digital creator, education, entrepreneur, health/beauty, editor, writer, personal blog, product/service, gamer, restaurant, beauty/cosmetics and personal care, food shop, photographer, retail sales and purchases, and lastly, video creator.

The categories of fashion and beauty are two of the areas that attract the most interest from the audience, generating deals and sponsors for the content creators. This is what makes it particularly interesting to investigate the audiences, according to Monge-Benito, Elorriaga-Illera and Olabarri-Fernández [34], in order to assess the trends across the different networks in terms of shopping recommendations and the audience that is most attracted to them.

Further research is needed to improve the ethical standards of media and media education to differentiate what is truthful and rigorous from what is mere entertainment, also aiming to avoid real-life harassment situations derived from the characters appearing in different television programmes, by learning to discern between reality and fiction, the character and the person [35]. It is along the lines of addressing the self-attribution by content creators to specific topics and trends (with or without ethical and objective foundations) that this study aims to focus on, verifying the real correlation between topic and content in the health influencers with the greatest global reach. Attempting to identify which content is taking advantage of health’s potential to reach and influence more audiences is of great interest. This is necessary to raise awareness about the ethics of the most relevant content creators, the control policies of social media platforms and the need for media education to provide defensive instruments to the youngest users who may be most conditioned by this type of opinion leader [36].

One of the main factors underlying this study is the demand for the Instagram influencers with the greatest impact on the population to disseminate accurate information true to the topics to which they are attributed, especially in sensitive areas such as politics and health. The repercussions of irresponsible practice by these popular content creators have a particular impact on young people, given their difficulties in identifying reliable information critically, their lack of open resources to obtain reliable information at the right time, and their unawareness of the relevance of digital applications in the field of health [30].

Our research focuses specifically on Instagram because it has been one of the fastest growing social networks in recent years [37,38,39,40].

There have been numerous approaches to this platform from the scientific literature in different fields. This service has been analysed as a pedagogical resource [41], as a journalistic [42] and marketing tool [43], as well as from the perspective of its business applications [44,45]. Given the nature of our work, it is especially interesting to review some of the main and most recent contributions to the study of Instagram and its intersection in the field of health [46]. In this sense, the work of Boulos et al. [47] investigated the strengths of this social network in health care from a general perspective. More specifically, Ellington et al. [48] analysed the effectiveness of using Instagram to recruit volunteers for research on diet, physical activity and obesity. The works of Yeung et al. [49] approached the study of this type of platform as spaces where abundant medical misinformation circulates. However, much of the work on this social network has focused on mental health [50]. Faelens et al. [51] studied the association between the use of this platform and diseases such as depression. Kim et al. [52] have focused their work on studying the effectiveness of this type of social media as a tool for detecting depressive behaviours. The impact of the use of networks such as Instagram on the mental health of adolescents has also been widely analysed [53]. Finally, authors such as Picardo et al. [54] have investigated the relationship between Instagram and suicidal behaviour.

In this context, the present study is based on three fundamental objectives:To describe the contents of self-proclaimed health influencers on Instagram and their authors, as well as to identify their main followers (age and gender profile).Determine the extent to which these accounts provide genuine health content to verify whether, as profiles that disseminate health information, they promote health to a high degree.Determine what factors (gender and age range of the influencer and predominant gender of their followers) influence the publication of health content on these accounts.

## 2. Method and Data Collection

To meet the aims of this study, a sample of Instagram influencer accounts in the health category with a high volume of followers was analysed. To select the sample, the Starngage platform was used, which provides an international ranking of theme-based influencers. The Starngage tool allowed for the identification of Instagram profiles classified under the health category. All the accounts with more than 1,000,000 followers were selected and processed. A comparative analysis of the results obtained by the tool was carried out, compiling records of the most-followed social media profiles on Instagram in 39 countries. Starngage defines itself as a “marketplace for content creators” that connects “brands with wonderful content creators from different channels and media”. They understand that “socially distributed visual content is the future of advertising” [55]. It is an interactive archive of the Instagram accounts with the most followers and engagement.

The platform has expanded to YouTube and TikTok over the past year; however, Starngage does not include a large volume of accounts nor the complete data from these two platforms as it is still in beta version for both of them, which is why its results have not been included in the present study.

When using the tool to narrow the search results to the health category alone, 67,375 global influencer profiles were found ranked from highest to lowest follower count. The search, conducted on 14 September 2022, yielded 66,542 influencer profiles from Instagram, TikTok and YouTube. After setting the minimum number of followers to 1,000,000, with no maximum number of followers, 443 entries were obtained that constitute the final sample.

Each Starngage profile provides a wide range of data about the owner of the profile and their interaction with the audience, among other details (Figure 1).

After selecting the sample, and in order to achieve the objectives described above, a quantitative content analysis research study was designed using descriptive and inferential statistics. The variables included in the study are presented in the Codebook (Table 1). These variables were quantitatively processed for each of the accounts that comprise the sample of this study: (1) gender of the influencer, (2) percentage of female followers, (3) percentage of male followers, (4) predominant age range of the followers (for this variable only two possible ranges have been considered, 18–24 and 25–34 years, since these are the ones provided by Starngage and also coincide with the interests of this study, focused on the younger population), (5) audience interests, in addition to health content, and (6) type of content. Keep in mind that in the “audience interests” variable, other topics of interest are categorized apart from “Health”, which is the predominant one among the followers of the selected accounts, according to Starngage. To determine the content typology of each account, 18 categories were established, which were coded and identified by analysing the last six publications of the profile in the corresponding social network.

Based on Objectives 1 and 2, the type of content posted by these accounts is one of the most important variables studied, for which it was determined to cross this variable with some of the sociodemographic characteristics of the influencers (gender) and their followers (predominant age range) obtained through Starngage. To analyse the type of content published by the influencers according to their gender, a contingency table analysis was conducted between the variables (“type of content” and “gender of the influencer”), with a chi-square test to observe significant differences in the type of content published depending on whether the influencer is male or female. The same statistical calculations were carried out to determine the type of content published (and the presence of significant differences) according to the predominant age range of the followers of each account. These data are presented in the Section 3.2.

To further explore the data, a bivariate correlational study was carried out between the different variables under study (Section 3.3.). Prior to this, and to determine whether to apply parametric (Pearson’s test) or non-parametric (Spearman’s test) calculations in the analysis of the correlations, the normality of the sample was verified using the Kolmogorov–Smirnov test, which found an absence of normality in the distribution of values in the sample, so it was decided to apply non-parametric tests (Spearman’s rho coefficient) for the correlational study.

The variables for which correlations were analysed are (1) “gender”, (2) “percentage of women following the account”, (3) “percentage of men following the account” and (4) “predominant age range of followers”. To these variables, a dichotomous dummy variable called “type of content” was added, coded as follows:Content not related to health. Groups together all categories of the “content” variable that do not refer to health-related aspects (e.g., brands, normative bodies, beautiful person, memes and virals, swimwear or underwear, etc.). Value assumed by this category of the variable: 0.Health-related content. Groups the following categories of the “content” variable: (1) healthy food, (2) unhealthy food, (3) health tips, (4) pregnancy, (5) sport and (6) hospital environment. Value assumed by this category: 1.

Once the variables that were statistically associated with the dependent variable “type of content” were detected, a multiple linear regression test was conducted to determine the factors that predict the behaviour of this dependent variable (Section 3.3), to achieve Objective 3.

All statistical calculations were performed with SPSS v. 26 software.

## 3. Results

### 3.1. General Data

Of the 443 accounts analysed, 66.4% (*n* = 294) belong to women and 27.1% (*n* = 120) to men. The gender of a total of 29 profiles (6.5%) could not be identified. There are more accounts with a predominant follower age range between 25 and 34 (*n* = 278; 62.8%) than those followed by users aged between 18 and 24 (*n* = 109; 24.6%). No data was available on the follower age range for a total of 56 profiles (12.6%). It should be noted that only two age ranges were included (18–24 and 25–34). As for the gender of the users, the sample analysed was very balanced, with 50.4% of the followers of the profiles studied being male, and 49.6% being female. The users of these profiles are mostly interested in the beauty category (*n* = 213; 48.1%), with fitness (*n* = 57; 12.9%) and entertainment (*n* = 26; 5.9%) as the second and third most prevalent topics (Table 2).

### 3.2. Content Analysis

Health-related content in the accounts observed is minimal (Table 3). The influencers studied included references to healthy foods in only 1.5% of their posts (*n* = 21). They posted about unhealthy foods in 1.3% of occasions (*n* = 18). Health advice was present in 2.2% of publications (*n* = 31). Sport is mentioned in 5.4% (*n* = 75), pregnancy in 0.6% (only in eight publications) and hospital environment in 0.3% (four publications). However, content about normative bodies (172; 12.3%), brands (*n* = 163; 11.7%), swimwear or underwear images (*n* = 139; 9.9%) or beauty (*n* = 139; 9.9%) was more frequent.

Hypothesis testing using chi-square tests found highly significant differences in the distribution of content by gender (X^2^ = (19, *n* = 1398) = 170.574, *p* < 0.001). Female influencers share more posts about healthy food than male influencers (Table 4), as 1.5% of the content posted by female influencers falls into this category compared to 0.7% for male influencers. However, male influencers publish more unhealthy food references (2.1% compared to 0.5% for female influencers). A greater deviation is observed regarding health tips, a category covered in 4.2% of men’s posts and only 1.5% of women’s posts. Men also incorporate more sport content (10.6%), representing 6.2% of the content posted by women.

For content not related to health, there are notable gender differences in the “brands” category (predominantly posted by men: 23.2% compared to 12.4% by women), “normative bodies” (more content shared by female accounts: 19.2% compared to 10.9% by male accounts) and “beautiful person” (higher proportion shared by female accounts: 16.8% compared to 5.6% by male accounts).

This section may be divided by subheadings. It should provide a concise and precise description of the experimental results, their interpretation, as well as the experimental conclusions that can be drawn.

There are also very significant differences in terms of content distribution of the influencers analysed according to the age of their followers (X^2^ = (19, *N* = 1398) = 35.127, *p* < 0.013). Overall, a greater interest in health topics is observed in those accounts with a majority of followers between the ages of 25 and 34, which is the oldest age range analysed in this study (Table 5). Influencers with a higher number of followers in the 25–34 age range share more content about unhealthy food (2.5%) than those whose followers are between 18 and 24 (1.2%). They also give more health advice (3.4% vs. 0.8%) and post almost twice as much content about sports (8.5% vs. 4.4%). The only health-related category favoured by accounts followed by the younger age group is “pregnancy”, representing 0.8% of the content from accounts followed mainly by users between the ages of 18 and 24, compared to 0.6% from accounts with a majority of followers aged between 25 and 34.

### 3.3. Correlational Study and Predictive Factors of the Type of Content

Table 6 shows the complete correlational study of variables, as explained in the Method and Data collection section. The following data are relevant:Highly significant correlations are observed in a positive sense (although with weak intensity) between gender and the presence of health content (rho(1398) = 0.102 *p* = 0.001). In this case, the fact that the influencer is male is significantly associated with the publication of health-related content.Again, the calculations also find a highly significant positive (but equally weak) correlation between accounts with a majority of followers aged between 25 and 34 and the publication of health-related content (rho(1398) = 0.120 *p* < 0.001).

Given the existence of variables (gender and age range) that correlate in a statistically significant way with the posting of health-related content, a multiple linear regression test was performed in order to determine to what degree these independent variables (gender and age range) predict the behaviour of the dependent variable (type of content) (Study Objective 3). To this end, a stepwise linear regression test was conducted, which confirmed that the two independent variables are predictors of health-related content in the accounts analysed (Table 7). The factor that best predicts the posting of health content is the predominant age range of the followers (β = 0.125, *p* < 0.001), before the gender of the influencer (β = 0.101, *p* = 0.003). However, it is observed that, based on the last step, the model has a low predictive capacity for the behaviour of the “type of content” variable (R^2^ = 0.024; R^2^ adjusted = 0.022) (R = 0.156, F(1, 1398) = 10.88), *p* < 0.001. The model predicts the publication of health-related content by 2.2%.

## 4. Discussion

This study contributes to the discussion around the activity of influencers who publish health content on Instagram. In this field, the scientific literature has focused on the impact of this type of content regarding disorders such as anorexia [56], sexual and reproductive health [57], mental health [58] or eating habits [59]. The latter study demonstrated the difficulty of controlling adolescents’ exposure to negative nutritional health messages. Overall, these studies document the strong impact that influencers have on the health habits of their audience, especially among younger generations, the population group targeted in this study.

In accordance with studies such as that of Hendry et al. [60], the present findings suggest that influencers who post health-related content on Instagram also display a highly superficial lifestyle and values that can sometimes confuse younger people about the true benefits of staying healthy. This study shows that the influencers analysed publish fewer health posts (and more about beauty and normative bodies) when the predominant age range of their followers is lower. However, both messages (beauty and health-related) are found together in the same accounts, which may reinforce the association between the idea of a healthy body and a normative, muscular body. These aspects can be problematic, as studies such as Durau et al. [61] reported an association between the physical attractiveness of the influencer who publishes health content and the degree of trust among their followers. It is therefore essential that platforms make an effort to clearly differentiate between both types of content: (1) genuinely health-related content and (2) posts where health is only a secondary aspect of having a beautiful body, which young people may indirectly perceive as a healthy physique. Given this circumstance, digital platforms should improve the control mechanisms for theme-based profiles with the highest engagement and number of followers, avoiding, as far as possible, intrusive content interfering with sensitive topics such as health. Future research lines are proposed to further examine these disinformation practices in other knowledge areas, such as education on different platforms, especially on YouTube and Instagram. It would be advisable to examine the rigour and reliability that followers of these areas perceive from the accounts that belong to them, and to evaluate these types of classifications in order to improve the user experience on social networks. While it is not a direct objective of this study, it would be useful to delve into the algorithms that manage the content on Instagram or other platforms to understand the effect and direct implication that the network itself has on the thematic classification and what users can or cannot expect from it. There are certain categories, such as education or health, which may be of interest to users. As such, those profiles with a significant number of followers should be supervised to ensure that the content shared corresponds to the category assigned by the influencers themselves.

Despite not being predominant in their accounts, certain influencers tagged under the health category may provide incorrect information on this topic when they refer to health issues which, as shown in this study, are usually addressed from a frivolous point of view or characterised by spectacularisation [62]. The problem is aggravated by the high credibility that certain population groups attribute to major influencers, even if the content they publish is not related to their professional field. In this sense, previous studies found that popular influencers who publish health content (analysed in this study) “have greater recognition and repercussion than the profiles of health professionals and experts” [63]. Such credibility in the health content published by non-specialist accounts is a consequence of the high degree of distrust that political, media and health institutions have aroused among citizens in recent years, which has led the public to seek other references through social networks and instant messaging services such as WhatsApp [64].

On the other hand, while this study did not aim to address the aesthetic–normative and sexual content in these types of accounts on Instagram, this parameter was present in the analysis, and the data show a notable difference between the prevalence of content on normative bodies, beauty or eroticism published in male (17.9%) and female (44.5%) influencer profiles. This aspect should lead to further analysis in future studies.

This study is not without a number of limitations that should be noted. Firstly, data retrieval from the accounts analysed relies on the accuracy of a platform such as Starngage, which could have made certain errors when categorising users in the health category. The tool used could also present certain geographical biases by focusing on influencers from certain regions, especially those from English-speaking countries, and thereby gathering more data on them. However, these problems are common in social science research when using specific software for social network data mining.

Another limitation has to do with using Instagram as the main network for the study. Unlike other platforms such as TikTok, which only use one media format (video), content on Instagram can be presented in different formats (photo, photo carousel and video), which could lead to differences in the content published if only one format had been considered (e.g., only video). Moreover, most of the content of influencers on Instagram is published not as a fixed post, but in the form of an ephemeral story that lasts only 24 h, meaning that a large volume of messages that had an impact on followers were not included in the study due to the impossibility of collecting them due to their brevity. Given that the success of influencers is mainly due to the use of video and a familiar and non-technical language [65,66], future research could focus exclusively on the study of influencer health content in video format, to identify possible differences with the results observed in samples that integrate different visual formats.

## 5. Conclusions

This study examined the 443 accounts tagged under the health category with the largest global reach on Instagram. In 90% of cases, these accounts did not publish any health-related content. Most of the followers of the accounts that allude to a healthy lifestyle are between 25 and 34 years old, and these profiles are mainly of male users.

As for the rest of the content included within the health category but unrelated to it, there is a prevalence of brands, especially in the profiles of male users and of normative bodies among female Instagrammers.

Although health tips are more often posted by women, the other topics related to healthy living are, in general, published more often by men. Together with the fact that the majority of followers belong to the 25–34 age group, this is a predictive factor for health-related content on Instagram. Indirectly, this finding offers the opportunity to investigate more content variables, while also assessing the complexity of this parameter and its limited contribution as a predictive factor.

The lack of ethical responsibility in social networks can be particularly conflictive when it comes to areas such as health, given the growing trend of consuming, consulting and referring to these platforms among the population. By identifying accounts with a high reach in the field of health, and analysing their content, it becomes clear that the publications of these accounts do not coincide with healthcare guidelines or habits, but rather feature images and videos that promote beauty, normative bodies and advertising. Young people are increasingly becoming both senders and receivers of online content and, interacting on social networks is the second most frequent activity during the average three hours that a child between 9- and 16-years-old spends on the Internet daily in Spain, excluding educational activities [4].

Finally, it is worth questioning the transcendence of some of the results of this study, especially the widespread and indiscriminate consumption of content about brands and normative bodies. What effects are these messages having on the development of our young people’s identities? Educommunication experts have already acknowledged the importance and influence of this type of messages for the establishment of social media and for the kind of referents that children [67] and young people are adopting as a result of the regular consumption of content with such commercial manifestations. Programmes that adapt educommunication in basic education and further develop a critical attitude towards this type of content could be of particular interest, including an evolutionary approach that allows to examine how the implementation of existing educommunication curricula, such as UNESCO’s Media and Information Literacy [68], affects the critical education of the younger population. Undoubtedly, educational policies should include educommunication programmes that examine these aspects in depth and allow users to interact freely and critically with social networks.

## Figures and Tables

**Figure 1 ijerph-19-15817-f001:**
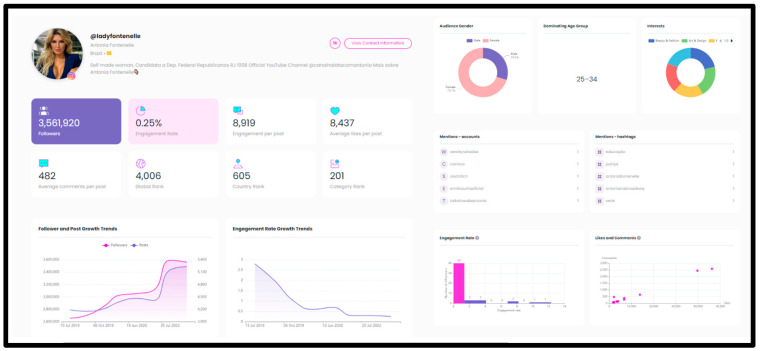
Instagram profile of @ladyfontenelle on Starngage (retrieved 4 October 2022). Source: Starngage.com.

**Table 1 ijerph-19-15817-t001:** Codebook. Source: own elaboration.

Variable	Category
1. Influencer gender	Male Female
2. % female followers	Quantitative variable
3. % male followers	Quantitative variable
4. Follower’s interests *	Beauty Music Fitness Travel Business Art and design Children and family Cars Entertainment Food and restaurants Movies and TV Sports Photography Not identified
5. Follower age range	18–24 years 25–34 years
6. Content published by the influencer	Brands Normative body Healthy foods Unhealthy foods Health tips Swimwear or underwear Eroticism Beautiful person Non-normative body Family Pregnancy Sport Beauty tips Memes and virals Bodybuilding Hospital environment Cooking recipes Other

* in addition to health content.

**Table 2 ijerph-19-15817-t002:** Other interests of the followers analysed. Source: Own elaboration.

Other Follower Interests	*n*	%
Beauty	213	48.1
Music	18	4.1
Fitness	57	12.9
Travel	7	1.6
Business	1	0.2
Art and design	12	2.7
Children and family	13	2.9
Cars	2	0.5
Entertainment	26	5.9
Food and restaurants	18	4.1
Movies and TV	12	2.7
Sports	2	0.5
Photography	1	0.2
Not identified	60	13.5

**Table 3 ijerph-19-15817-t003:** Content published by the influencers analysed. Source: own elaboration.

Content	*n*	%
Brands	163	11.7
Normative body	172	12.3
Healthy foods *	21	1.5
Unhealthy foods *	18	1.3
Health tips *	31	2.2
Swimwear or underwear	139	9.9
Eroticism	67	4.8
Beautiful person	1	0.1
Non-normative body	17	1.2
Family	52	3.7
Pregnancy *	8	0.6
Sport *	75	5.4
Beauty tips	16	1.1
Memes and virals	10	0.7
Bodybuilding	25	1.8
Hospital environment *	4	0.3
Cooking recipes	12	0.9
Other	429	30.7

* health-related content.

**Table 4 ijerph-19-15817-t004:** Type of content published based on the influencer’s gender. Source: own elaboration.

Content	Female **	Male **
Brands	90 (12.4%)	66 (23.2%)
Normative body	140 (19.2%)	31 (10.9%)
Healthy foods *	11 (1.5%)	2 (0.7%)
Unhealthy foods *	4 (0.5%)	6 (2.1%)
Health tips *	11 (1.5%)	12 (4.2%)
Swimwear or underwear	116 (15.9%)	22 (7.7%)
Eroticism	62 (8.5%)	4 (1.4%)
Beautiful person	122 (16.8%)	16 (5.6%)
Non-normative body	16 (2.2%)	1 (0.4%)
Family	40 (5.5%)	12 (4.2%)
Pregnancy *	8 (1.1%)	0
Sport *	45 (6.2%)	30 (10.6%)
Beauty tips	9 (1.2%)	3 (1.1%)
Memes and virals	4 (0.5%)	4 (1.4%)
Bodybuilding	9 (1.2%)	16 (5.6%)
Hospital environment *	1 (0.1%)	3 (1.1%)
Cooking recipes	7 (1%)	2 (0.7%)

* health-related content; ** the percentage refers to the prevalence of content posted for each category according to the gender of the influencers.

**Table 5 ijerph-19-15817-t005:** Type of content published according to the predominant age range of the influencer’s followers. Source: own elaboration.

Content	18–24 **	25–34 **
Brands	36 (14.5%)	105 (15.4%)
Normative body	49 (19.8%)	102 (15%)
Healthy foods *	3 (1.2%)	17 (2.5%)
Unhealthy foods *	2 (0.8%)	16 (2.4%)
Health tips *	2 (0.8%)	23 (3.4%)
Swimwear or underwear	32 (12.9%)	91 (13.4%)
Eroticism	10 (4%)	46 (6.8%)
Beautiful person	42 (16.9%)	85 (12.5%)
Non-normative body	4 (1.6%)	11 (1.6%)
Family	13 (5.2%)	30 (4.4%)
Pregnancy *	2 (0.8%)	4 (0.6%)
Sport *	11 (4.4%)	58 (8.5%)
Beauty tips	5 (2%)	7 (1%)
Memes and virals	2 (0.8%)	4 (0.6%)
Bodybuilding	2 (0.8%)	19 (2.8%)
Hospital environment *	0	3 (0.4%)
Cooking recipes	2 (0.8%)	10 (1.5%)

* health-related content; ** the percentage refers to the prevalence of content posted for each category by the influencers according to the predominant age range of their followers.

**Table 6 ijerph-19-15817-t006:** Correlational study. Source: own elaboration.

	Gender (*p*)	% Females (*p*)	%Males (*p*)	Age Range (*p*)	Type of Content (Dummy) (*p*)
**Gender (*p*)**		−0.059 (0.079)	0.059 (0.080)	−0.065 (0.052)	0.102 (<0.001)
**% Females (*p*)**			−1.000 (0.000)	−0.327 (0 < 0.001)	0.059 (0.073)
**%Males (*p*)**				0.328 (0 < 0.001)	−0.059 (0.071)
**Age range (*p*)**					0.120 (<0.001)
**Type of content**					

**Table 7 ijerph-19-15817-t007:** Predictor variables for type of content. Source: own elaboration.

Step	Predictor Variable	Rho Spearman	Standardised Coefficient	*p*
1	Age range	0.120 *	0.119	<0.001
2	Age range		0.125	<0.001
	Gender	0.102 *	0.101	0.003
**Model summary (last step)**				
**F**		** *p* **	**R**	**R^2^ (R^2^ adjusted)**
10.88		<0.001	0.156	0.024 (0.022)

Significance of correlation coefficients (rho): * *p* < 0.001.

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
