# Peer review of "Health Promotion on Instagram: Descriptive–Correlational Study and Predictive Factors of Influencers’ Content"

_ijerph, 2022, doi:10.3390/ijerph192315817_

Round 1

Reviewer 1 Report

Thank you for allowing me to review this contribution entitled

Health promotion on Instagram. Descriptive-correlational study and predictive factors of influencers’ content

In general, the topic of the paper is interesting but there are fundamental aspects that should be reviewed in order to add value. It is difficult to understand the focus of the study (population, context). As well as the choice or maintenance of the category "health" when the interests mentioned by the followers do not include health issues, and the publication on health aspects is very much in the minority by the influencers. I understand that the problem may lie in the classification made by the tool to include the list of influencers but not so much in whether they themselves or their followers identify with the topic of health. 

Format

The article should be adapted to the recommended section structure of the journal.: Introduction, Materials and Methods, Results, Discussion, Conclusions.

Introduction

L31 Modify the sentence “According to data provided by Ref.[2]” The author or institution should be included. Idem for L39, L66, L74, L77 Please, review the whole document 

The sentence L34 to L38 is difficult to understand

L 47 There is a 3 which seems to be an error or I don't know what it means.

In general the introduction is too long and not focused.  At the beginning it seems that the authors are going to focus on children or adolescents, but then it seems that they are going to focus on the general population. It is also difficult to guess whether the authors want to deal with information from the internet, from social networks, from TV commercials…

Materials and Methods

L172 The authors say  that  an instrument was used, Is it Starngage tool that instrument? Please clarify

L187  A Figure 2 is mentioned but is not included

Is the focus category health or health and fitness?

It is not clear why influencers are chosen from all over the world and in all media, and in the introduction it is quite focused on Spain. 

The category "health" is not present among the interests of the followers,  isn't it strange?  In fact in the following section it is mentioned that Beauty is the main topic of interest. 

Results

The statistical analyses carried out should be mentioned in the methodology section. 

Discussion

As the authors say It should be noted that only two age ranges were included (18-24 and 25-34)  and the majority of followers belong to the 25-34 age group. However many of the observations focus on the child or adolescent population and on the Spanish context when the accounts analysed were international. 

Study limitations have not been included

Author Response

The response to the comments of reviewer number 1 has been integrated in the attached file, however, if it is not correctly attached, it is included in the table below.

Health promotion on Instagram. Descriptive-correlational study and predictive factors of influencers’ content

Reviewer

Suggested modification

Answer

1

It is difficult to understand the focus of the study (population, context).

The explanation of the research focus has been corrected, identifying the object of study in a more concrete and coherent way, in accordance with the state of the art, the working hypotheses and the method. These modifications can be seen notably in sections 1 and 1.1.

As well as the choice or maintenance of the category "health" when the interests mentioned by the followers do not include health issues, and the publication on health aspects is very much in the minority by the influencers. I understand that the problem may lie in the classification made by the tool to include the list of influencers but not so much in whether they themselves or their followers identify with the topic of health. 

In the first sections, special emphasis has been placed on the importance of social media content for younger users, justifying the need to address, among said content, those that can have a more negative influence on young people lacking media education, such as are the contents referring to the field of health. In point 1.1 can be seen the corrections applied in this regard, following the recommendations of the reviewer.

Format. The article should be adapted to the recommended section structure of the journal.: Introduction, Materials and Methods, Results, Discussion, Conclusions.

The structure of the article has been adapted to the organization and sections recommended by the journal, adding the Discussion section as a separate heading from the Conclusions. Now, the section structure in the article is:

1.        Introduction and literature review

2.        Method and data collection

3.        Results

4.        Discussion

5.        Conclusions

L31 Modify the sentence “According to data provided by Ref.[2]” The author or institution should be included. Idem for L39, L66, L74, L77 Please, review the whole document 

Throughout the document, the terms "Ref" have been eliminated, conveniently identifying the authorship of the referenced source.

The sentence L34 to L38 is difficult to understand

The sentence has been corrected and changed.

L47 There is a 3 which seems to be an error or I don't know what it means.

The sentence has been modified for better understanding and the writing of the numbers has been corrected. The data on youth screen consumption has been expanded, extending to the international arena, in line with the modifications and corrections proposed by the reviewer.

In general the introduction is too long and not focused.  At the beginning it seems that the authors are going to focus on children or adolescents, but then it seems that they are going to focus on the general population. It is also difficult to guess whether the authors want to deal with information from the internet, from social networks, from TV commercials…

The explanation of the research focus has been corrected, identifying the object of study in a more concrete and coherent way, in accordance with the state of the art, the working hypotheses and the method.

In the first sections, special emphasis has been placed on the importance of social media content for younger users, justifying the need to address, among said content, those that can have a more negative influence on young people lacking media education, such as are the contents referring to the field of health. Please, see sections 1 and 1.1.

L172 The authors say that an instrument was used, Is it Starngage tool that instrument? Please clarify.

It has been clarified on page 4 (at the beginning of the Method and data collection section).

L187  A Figure 2 is mentioned but is not included.

The reference to the non-existent Figure 2 has been removed.

Is the focus category health or health and fitness?

In the section dedicated to the methodology and in the conclusions, it is clarified that the study focuses on the accounts labeled under the category of health.

It is not clear why influencers are chosen from all over the world and in all media, and in the introduction it is quite focused on Spain. 

The specific references to Spain have been modified and expanded with international data, representative of the scope of the study and its method. Please, see Section 1.

The statistical analyses carried out should be mentioned in the methodology section. 

The section on methodology has been extensively reinforced in order to include a description of the statistical analyzes carried out. In addition to adding explanations not present in the original version, the description of the correlational analysis is introduced in the methodological section, which in the first manuscript was at the beginning of section 3.3. As stated by the reviewer, the explanation of the correlational analysis is better placed in the methodological part, not in the results section.

Discussion. As the authors say It should be noted that only two age ranges were included (18-24 and 25-34)  and the majority of followers belong to the 25-34 age group. However many of the observations focus on the child or adolescent population and on the Spanish context when the accounts analysed were international. Study limitations have not been included.

The discussion has been totally restructured, including numerous references and expanding the exposure of the limitations of the study and other observations of the team of researchers who have carried it out, supported by recent studies. Please, see section 4.

Reviewer 2 Report

I would like to thank the editors for the opportunity they have given me to review this article. I would also like to congratulate the authors for their work and effort.

The article seems to me to be very interesting, of current relevance. It is within the scope of the journal. The dissemination of health-related content by influencers is a particularly sensitive issue for society, especially because of the possible consequences of their recommendations (many of them made by non-health professionals).

Having said that, I will try to make a number of recommendations that could improve the final result of the work:

1) I consider the abstract to be well structured, highlighting the objective and advancing some of the results of the research.

2) In terms of the introduction and literature review, it is also well developed and fulfils its function. However, I believe that the research gap covered by this work should be highlighted, for example, by reinforcing the considerations made by the authors between lines 152 to 160. I believe that the authors should do so in section "1.1. Social networks and health", making a brief reflection on the risk derived from the recommendations made by influencers, for example:

Cabeza-Ramírez, L. J., Sánchez-Cañizares, S. M., Santos-Roldán, L. M., & Fuentes-García, F. J. (2022). Impact of the perceived risk in influencers' product recommendations on their followers' purchase attitudes and intention. Technological Forecasting and Social Change, 184, 121997.

And in the specific field of health:

Borah, P., & Xiao, X. (2018). The importance of 'likes': The interplay of message framing, source, and social endorsement on credibility perceptions of health information on Facebook. Journal of health communication, 23(4), 399-411.

One paragraph would suffice, and I think this might help the reader to understand the importance of the research presented. In a context where recommendations made by certain (non-professional) influencers could affect the health of their followers.

3) Another aspect that I think would improve the flow of the article would be to mark the objectives of the article (as the authors have done), but leave them at the end of the introduction. In an article for a scientific journal, it is strange to dedicate a section to the objectives.  On the other hand, the methodology does need such a section.

 4) I think that the methodology section should be restructured and renamed "methodology and data collection": and first explain the platform used, "Starngage", and how the data were collected (this appears in the section). Then explain the statistical techniques to be used and justify why they are appropriate. Statistical correlation analysis may be very basic, but it helps to shed light on the links between variables involved in new phenomena (such as the one presented here).

5) If you present the methodology as I have recommended, the presentation of the results makes more sense (as you have done it). I have not observed significant errors in this section. Beyond punctuation marks, "the dot is used to express decimals, unlike in Spanish" (for example, in table 3).

6) Finally, the discussion and conclusions should be expanded. Authors should make an effort to relate the findings they have presented in the results section to previous literature. Showing those works that their results contribute to reinforce (or go in the same line), and possible divergences. Furthermore, I consider it vitally important for a scientific article to include in this section a subsection on "practical implications", and a subsection on "limitations and future lines of research" as a priority.

In summary, my recommendation is to accept the paper by introducing major/minor revisions. I congratulate the authors for their work, but I think they should strengthen the paper by following the standards for publication in a scientific journal. I look forward to reading the final version of the text soon.

Author Response

The response to the comments of reviewer number 2 has been integrated in the attached file, however, if it is not correctly attached, it is included in the table below.

Health promotion on Instagram. Descriptive-correlational study and predictive factors of influencers’ content

Reviewer

Suggested modification

Answer

2

I consider the abstract to be well structured, highlighting the objective and advancing some of the results of the research

The abstract refers to the objective of our research, to find out what self-styled health content is like, to identify predictive factors in the consumption and monitoring of this type of account, and to describe its followers: “This study aims to describe the content of these profiles and their authors and to determine whether they promote health as accounts that disseminate health-related content, identifying predictive factors of their content topics. In addition, it aims to portray their followers and establish correlations between the gender of the self-described health influencers, the characteristics of their audience and the messages that these prescribers share”.

In it, we reflect that the results of the study confirm that self-proclaimed health content does not promote it, and establishes the predictive pattern of followers and influencers in this content área: “Health promotion is not the predominant narrative among these influencers, who tend to promote beauty and normative bodies over health matters. A correlation is observed between posting health content, the gender of the influencers and the average age of their audiences”.

I believe that the research gap covered by this work should be highlighted, for example, by reinforcing the considerations made by the authors between lines 152 to 160. I believe that the authors should do so in section "1.1. Social networks and health", making a brief reflection on the risk derived from the recommendations made by influencers, for example:

Cabeza-Ramírez, L. J., Sánchez-Cañizares, S. M., Santos-Roldán, L. M., & Fuentes-García, F. J. (2022). Impact of the perceived risk in influencers' product recommendations on their followers' purchase attitudes and intention. Technological Forecasting and Social Change, 184, 121997.

And in the specific field of health:

Borah, P., & Xiao, X. (2018). The importance of 'likes': The interplay of message framing, source, and social endorsement on credibility perceptions of health information on Facebook. Journal of health communication, 23(4), 399-411.

One paragraph would suffice, and I think this might help the reader to understand the importance of the research presented. In a context where recommendations made by certain (non-professional) influencers could affect the health of their followers.

This recommendation has been taken into account and emphasis has been placed on this approach suggested by the reviewer, and references to the sources proposed by the reviewer have been incorporated. An explanatory paragraph has been expanded to enhance the meaning, need and coverage that our work provides and that pursues its approach. Please, see section 1.1

Another aspect that I think would improve the flow of the article would be to mark the objectives of the article (as the authors have done), but leave them at the end of the introduction. 

As suggested by the reviewer, the objectives of the work are introduced at the end of the introductory section, instead of at the beginning of the methodological section.

I think that the methodology section should be restructured and renamed "methodology and data collection": and first explain the platform used, "Starngage", and how the data were collected (this appears in the section). Then explain the statistical techniques to be used and justify why they are appropriate. Statistical correlation analysis may be very basic, but it helps to shed light on the links between variables involved in new phenomena (such as the one presented here).

In accordance with the suggestion, the title of the section has been changed. The recommended structure has been followed, relating the statistical tests carried out with the objectives of the research.

The discussion and conclusions should be expanded. Authors should make an effort to relate the findings they have presented in the results section to previous literature. Showing those works that their results contribute to reinforce (or go in the same line), and possible divergences. Furthermore, I consider it vitally important for a scientific article to include in this section a subsection on "practical implications", and a subsection on "limitations and future lines of research" as a priority.

A specific discussion section is introduced in order to contrast our results with those obtained by other authors (a total of 11 new references are added to the original manuscript).

Information on the limitations of the work is added and future lines of research are proposed.

Please, see sections 4 and 5.

Reviewer 3 Report

The content of the article is interesting, but there are many things that need to be improved.

Instagram is a keyword, but it is only in one sentence of the introduction. The introduction should be extended to include literature on this issue.

The study was insufficiently described and characterized. It described the content analysis very poorly.

The meaning of the important term is not sufficiently explained - health accounts.

Author Response

The response to the comments of reviewer number 3 has been integrated in the attached file, however, if it is not correctly attached, it is included in the table below.

Health promotion on Instagram. Descriptive-correlational study and predictive factors of influencers’ content

Reviewer

Suggested modification

Answer

3

Instagram is a keyword, but it is only in one sentence of the introduction.  The introduction should be extended to include literature on this issue.

Indeed, the use of Instagram as a keyword was problematic. Therefore, it has been removed as a keyword from the job.

The study was insufficiently described and characterized. It described the content analysis very poorly.

The section on methodology has been extensively reinforced in order to include a description of the statistical analyzes carried out. In addition to adding explanations not present in the original version, the description of the correlational analysis is introduced in the methodological section, which in the first manuscript was at the beginning of section 3.3.

The meaning of the important term is not sufficiently explained - health accounts

The concept of "health accounts" in the framework of our work is problematic, therefore all reference to that term is eliminated (it appeared only twice in the discussion and conclusions section).

Round 2

Reviewer 1 Report

Thank you for the changes made to the work. While I think they have improved the paper there are still some issues that I think should be reviewe

The introduction is still too long. They should try to synthesise and focus on the main aspects related to the objectives. 

How do you explain the fact that the health category does not appear among the interests of the followers?  (Table 1 and 2) This is something that should be brought up in the discussion. 

Author Response

Reviewer

Suggested modification

Answer

1

The introduction is still too long. They should try to synthesise and focus on the main aspects related to the objectives. 

A paragraph reviewing the scientific literature on Instagram and its relationship with health issues is introduced. In this way, Instagram is built as an object of study in our research and its choice as a platform to carry out our work is justified. This brief review of the literature on Instagram, which integrates 18 new references, is based on two criteria: (1) the thematic relevance in connection with our object of study and (2) the novelty of the studies, since the works are prioritized. published in recent years.

 With the modifications applied, the introduction has a total of 1970 words. Please consider a standard extension, taking into account that it includes an introduction, literature review and objectives.

How do you explain the fact that the health category does not appear among the interests of the followers?  (Table 1 and 2) This is something that should be brought up in the discussion. 

The question of audience interests is explained in the definition of the variables (methodology section, page 6/16). The topics of interest of the followers are categorized in addition to the health issues. This is also indicated in the tables where the results on this variable are exposed. The health category does not appear in this tables because it is already selected when screening the profiles for that kind of accounts. In other words, the table indicates that followers with interests in health are also interested in the following topics (…). For a better understanding, we have reinforced this clarification in the text of the Methodology section and in the tables.

Reviewer 3 Report

The word - Instagram is in the title so you shouldn't eliminate it from your keywords. In the introduction, you should refer to this word.

Author Response

Reviewer

Suggested modification

Answer

3

The word - Instagram is in the title so you shouldn't eliminate it from your keywords. In the introduction, you should refer to this word.

The word Instagram has been included again as a keyword and a paragraph reviewing the scientific literature on Instagram and its relationship with health-related issues is introduced. In this way, Instagram is built as an object of study in our research and its choice as a platform to carry out our work is justified.
